# Evaluating Feature Selection Methods for Accurate Diagnosis of Diabetic Kidney Disease

**DOI:** 10.3390/biomedicines12122858

**Published:** 2024-12-16

**Authors:** Valeria Maeda-Gutiérrez, Carlos E. Galván-Tejada, Jorge I. Galván-Tejada, Miguel Cruz, José M. Celaya-Padilla, Hamurabi Gamboa-Rosales, Alejandra García-Hernández, Huizilopoztli Luna-García, Klinge Orlando Villalba-Condori

**Affiliations:** 1Unidad Académica de Ingeniería Eléctrica, Universidad Autónoma de Zacatecas, Jardín Juárez 147, Centro, Zacatecas 98000, Mexico; valeria.maeda@uaz.edu.mx (V.M.-G.); gatejo@uaz.edu.mx (J.I.G.-T.); jose.celaya@uaz.edu.mx (J.M.C.-P.); hamurabigr@uaz.edu.mx (H.G.-R.); alegarcia@uaz.edu.mx (A.G.-H.); hlugar@uaz.edu.mx (H.L.-G.); 2Unidad de Investigación Médica en Bioquímica, Centro Médico Nacional Siglo XXI, Hospital de Especialidades, Instituto Mexicano del Seguro Social, Av. Cuauhtémoc 330, Col. Doctores, Del. Cuauhtémoc, Ciudad de Mexico 06720, Mexico; miguel.cruzlo@imss.gob.mx; 3Virrectorado de Investigación, Universidad Nacional Pedro Henríquez Ureña, Santo Domingo 1423, Dominican Republic; kvillalba@unphu.edu.do

**Keywords:** diabetic kidney disease, feature selection algorithms, risk factors, machine learning, random forest

## Abstract

**Background/Objectives**: The increase in patients with type 2 diabetes, coupled with the development of complications caused by the same disease is an alarming aspect for the health sector. One of the main complications of diabetes is nephropathy, which is also the main cause of kidney failure. Once diagnosed, in Mexican patients the kidney damage is already highly compromised, which is why acting preventively is extremely important. The aim of this research is to compare distinct methodologies of feature selection to identify discriminant risk factors that may be beneficial for early treatment, and prevention. **Methods**: This study focused on evaluating a Mexican dataset collected from 22 patients containing 32 attributes. To reduce the dimensionality and choose the most important variables, four feature selection algorithms: Univariate, Boruta, Galgo, and Elastic net were implemented. After selecting suitable features detected by the methodologies, they are included in the random forest classifier, obtaining four models. **Results**: Galgo with Random Forest achieved the best performance with only three predictors, “creatinine”, “urea”, and “lipids treatment”. The model displayed a moderate classification performance with an area under the curve of 0.80 (±0.3535 SD), a sensitivity of 0.909, and specificity of 0.818. **Conclusions**: It is demonstrated that the proposed methodology has the potential to facilitate the prompt identification of nephropathy and non-nephropathy patients, and thereby could be used in the clinical area as a preliminary computer-aided diagnosis tool.

## 1. Introduction

Declared as a global health emergency, diabetes mellitus (DM), or currently referred to simply as diabetes as established by the International Diabetes Federation (IDF), affects more than 537 million people worldwide. The IDF has estimated that the number of subjects with DM will increase from 643 million in 2030 to 783 million in 2045 [1]. In this sense, the early detection of DM is extremely important; nevertheless, there is a high number of people living with undiagnosed DM, approximately 240 million people worldwide.

In Mexico, the number of individuals with DM is predicted to increase from 14.1 million in 2021 to 22.3 million by 2045, of whom more than 95% would have had type 2 diabetes (T2D) [2,3]; which is an endocrine disease, resulting from a pancreatic β cell dysfunction combined with insulin resistance. Moreover, poor glycemic control results in multiple complications, including the development of micro and macro-vascular diseases.

Diabetic kidney disease (DKD) is one of the most common microvascular complications of T2D [4]. Clinically, it is distinguished by persistently elevated levels of albumin in the urine, a progressive decline in the glomerular filtration rate, and hypertension, leading to end-stage renal disease (ESRD) [5]. The traditional understanding of DKD progression has been marked by an initial rise in urinary albumin excretion, followed by the development of significant albuminuria and a subsequent rapid deterioration in renal function. As a result, proteinuria has long been viewed as a crucial indicator of declining renal health. Nevertheless, this theoretical model has been challenged by evidence demonstrating that some patients with proteinuria are able to revert to normal albumin excretion levels, either spontaneously or through comprehensive risk management strategies. These findings have raised doubts about the reliability of microalbuminuria as a marker and the appropriate timing of interventions in the management of DKD [4,6].

The percentage of ESRD attributed to DM varies between 10% and 67% because its prevalence is up to 10 times higher in people with DM [7]. In addition, it is a major health burden on health systems; for the specific case of Mexico, the cost generated in 2018 by DKD was $11,763 million (MXN, ~577,296,820 USD) for ESRD resulting from T2D [8].

It is a fact that the presence of T2D together with the development of its comorbidities, the care needs, treatment options, and associated costs have a significant impact. The pathogenesis of DKD is complex and is still not fully understood, resulting in poor therapeutic outcomes [9,10]. Therefore, early prevention, detection and diagnosis of DKD is fundamental in order to prevent its progression [11]. Actually, machine learning (ML) and artificial intelligence (AI) have played an important and transformative role in the healthcare industry. ML techniques have been extensively applied in various aspects of clinical decision-making and patient care, including but not limited to diagnosis, risk assessment, prediction, and prognosis [12,13,14,15]. These advancements have significantly enhanced the efficiency and accuracy of medical processes, leading to improved patient outcomes and more personalized treatment strategies. Furthermore, ML algorithms have exhibited exceptional proficiency in analyzing the extensive datasets of intricate healthcare information, encompassing electronic health records (EHRs), medical imaging, and genomic data. This analytical capability allows for the extraction of significant information and patterns that were previously inaccessible to healthcare practitioners. The incorporation of ML into healthcare frameworks holds the promise of transforming the methodologies employed in disease diagnosis, treatment and management. Consequently, this integration has the potential to usher in a new era of more efficacious and streamlined healthcare delivery.

The primary objective of this study is to explore the performance of four popular feature selection methods, namely, the Univariate filter method, Boruta, Galgo, and Elastic net, to identify risk factors that may contribute to the diagnosis of DKD in a clinical Mexican dataset; to evaluate the subset of relevant variables Random Forest was used for classifying non-DKD and DKD patients.

### Related Work

Recent research endeavors have prioritized the identification of the risk factors associated with DM, to enhance diagnostic decision-making processes. These investigations have significantly contributed to the advancement of predictive modeling and risk assessment within the realm of DM. Noteworthy studies, exemplified by the innovative feature selection and classification model for heart disease prediction [16], have yielded novel insights into the intricate interplay of factors influencing DM diagnosis. Furthermore, the study of Nagarajan et al. [17] has facilitated refined data analysis and real-time monitoring, thereby bolstering DM management initiatives. Additionally, Chang et al. [18], have elucidated population-specific patterns and trends, offering tailored strategies for DM prevention and treatment. Moreover, the detection of DR through principal component analysis multi-label feature extraction and classification [19], has underscored the imperative of early detection and intervention in mitigating DM-related complications. Collectively, these studies underscore the pivotal role of harnessing ML techniques and innovative methodologies in addressing the multifaceted challenges inherent in DM diagnosis. This comprehensive body of research serves as a foundation for understanding the complexities surrounding DM and sets the stage for exploring related conditions such as DKD.

In the case of Rodriguez-Romero et al. [20], data mining and ML techniques were applied to identify DKD biomarkers. The diabetic dataset consisted of 10,251 subjects with 18 factors. The identification of the most important features was performed using the InfoGain method. Then, six learning algorithms were tested (one rule [1R], J48 decision tree [J48], random forest [RF], simple logistic [SL], sequential minimal optimization [SMO], and naïve Bayes [NB]). Based on the experimental results, RF and SL exhibited the best performance.

Jiang et al. [21] proposed the prediction of DKD in T2D patients. A total of 302 Chinese subjects, and 19 clinical features were included. Focused on identifying a set of the most important features, a Least Absolute Shrinkage and Selection Operator (LASSO) analysis was conducted, where nine potential attributes were found. Then, a multivariable logistic regression model was performed with the candidate predictors. Finally, the concordance index (C-index) was calculated, obtaining 0.934.

Furthermore, Shi et al. [22] developed a DKD or diabetic retinopathy (DR) incidence risk nomogram. It is important to mention, that 4219 patients were divided into groups: T2D with DKD or DR, with both, and without any complication. LASSO regression was selected as a feature selection method, thus 7 of 23 attributes were potential predictors of DKD, and four in DR. To tackle this, a logistic regression analysis was conducted to optimize the feature selection; the risk factors with a *p*-value less than 0.05 were selected and combined to generate a prediction model. The performance of the multivariable logistic regression model showed an AUC of 0.807.

Likewise, Xi et al. [23], constructed a DKD risk nomogram; the study enrolled 1095 subjects divided into DKD (203) and non-diabetic kidney disease (NDKD, 892). With regard to reducing the predictors of DKD, a total of 23 clinical features were submitted into a LASSO regression model, obtaining 18 possible features. Then, from the remaining predictors, a logistic regression analysis was applied, resulting in 10 statistically significant predictors that were used to build the model. Their proposed nomogram prediction model achieved an AUC of 0.813, demonstrating good discrimination.

The investigation conducted by Maniruzzaman et al. [24] presented the comparison of machine learning (ML) methods for the prediction of DKD patients. The dataset consisted of 133 respondents having 73 cases, and 60 controls. The combination of principal component analysis (PCA) for feature selection, and six ML algorithms namely linear discriminant analysis (LDA), support vector machine (SVM), logistic regression (LR), k-nearest neighborhood (K-NN), naïve Bayes (NB) and artificial neural network (ANN) was implemented to select the best features at distinct PCA cutoff values. One of their experiments was to optimize the kernel of SVM demonstrating that SVM-radial basis function (RBF) provided better performance. Afterward, the best features fed into the six ML techniques, and it was noted that the SVM-RBF kernel obtained a maximum accuracy (88.7%), 0.91 AUC at a PCA cutoff of 0.96.

On the other hand, Yang and Jiang [25] developed a nomogram to assess the risk of DKD in T2D patients; 706 subjects and 23 features were included in the statistical analysis. A univariate logistic regression test was used to evaluate the significance of each variable; attributes with a *p*-value less than 0.05 were considered significant. Subsequently, 13 main features were subjected to a stepwise regression according to the principle of akaike information criterion (AIC), giving the best model with the smallest AIC value. A binary logistic regression model was constructed with the seven independent risk predictors; the prediction performance was measured using the C-index and AUC values. The results showed a 0.773 and 0.758 in the training and validation sets, respectively.

Table 1 summarizes the general approaches of the selected studies, including the feature selection algorithms, ML techniques, and the predictors of DKD. According to these relevant studies, we resolved to undertake a comprehensive comparison of different feature selection methods. These encompass a range of techniques, including filter-based (Univariate), penalty-based (Elastic net), tree-based (Boruta) and heuristic (GALGO) approaches.

Among these methods, Univariate evaluation ranks features based on a performance measure, and then selects the top variables with the highest scores [26]. An elastic net is designed to strike a delicate balance between accuracy and the magnitude of weight values within a linear model. It accomplishes this through the utilization of $L_{1}$ and $L_{2}$ regularization techniques. This is primarily attributed to its tendency to produce sparse models with minimal non-zero weight values [27]. Boruta, in comparison with other feature selection algorithms, adheres to an all-relevant variable selection approach, encompassing all features associated with the target output variable. In contrast, other techniques adhere to a minimum-optimum approach, relying on a limited subset of features to minimize errors in a chosen classifier. Furthermore, Boruta is adept at considering complex multi-variable relationships and can proficiently explore interactions between variables [28]. Lastly, GALGO, a genetic algorithm-based feature selection, initiates with a randomly generated population of models, subsequently refining these models through the emulation of natural selection mechanisms. This evolutionary process involves strategies such as an increased replication rate for the more efficient variable subsets, mutation for generating variations, and crossover for enhancing variable combinations. Significantly, the concurrent examination of variable sets organized within chromosomes during the selection process underscores the genuinely multivariate nature of variable selection [29]. The inclusion of these methods provides a holistic analysis of their performance in the context of DKD diagnosis in Mexican patients.

## 2. Materials and Methods

The architecture of the proposed ML classification system is shown in Figure 1, and has been discussed in the next subsections.

### 2.1. Dataset Description

In this study, the dataset was obtained from “Unidad de Investigación Médica en Bioquímica, Centro Nacional Siglo XXI, Instituto Mexicano del Seguro Social (IMSS)”. The Mexican patients signed an informed consent letter, and the protocol meets the Helsinki criteria which conforms to the ethical principles of the Ethics Committee of IMSS under the number R-2011-785-018.

Inclusion criteria for cases:-Patients diagnosed with T2D according to the ADA criteria: fasting glucose equal to or greater than 126 mg/dL;-Any gender;-Affiliated beneficiaries to the IMSS with active and up-to-date benefits at time of enrollment;-Not having any family members participating in the study;-Patients who agreed to participate in the study and signed the informed consent form;-Adults diagnosed before the age of 55, with the current age for inclusion being between 35 and 85 years.

Inclusion criteria for controls:-Individuals aged between 35 and 85 years, without meeting the criteria for metabolic syndrome (according to the International Adult Treatment Panel (ATP) III criteria);-Any gender;-Not having any family members participating in the study;-Individuals who agreed to participate in the study and signed the informed consent form;-Fasting glucose (12 h) below 100 mg/dL, and post-load glucose levels below 140 mg/dL two hours after ingesting 75 g of glucose.

Exclusion criteria:-Beneficiaries with temporary or seasonal insurance coverage-Individuals without a permanent residence of those who cannot be reached by phone, either at their home or through a family member;-Woman in the climacteric stage.

#### 2.1.1. Clinical Variables

Thirty-two baseline clinical variables were used as predictors to train and test the models: education (EDU), salary (SAL), sex, age, diagnosis age of diabetes (Age DX), waist hip ratio (WHR), body mass index (BMI), glucose (GLU), urea, creatinine (CRE), cholesterol (CHOL), high density lipoprotein (HDL), low density lipoprotein (LDL), triglycerides (TG), HDL uncorrected by treatment (UHDL), LDL uncorrected by treatment (ULDL), total cholesterol uncorrected by treatment (TCHOLU), TG uncorrected (UTG), systolic blood pressure (SBP), diastolic blood pressure (DBP), SBP uncorrected by treatment (USBP), DBP uncorrected by treatment (UDBP), hypertension treatment (HA-TX, refers to medical interventions designed to manage and control high blood pressure), lipids treatment (LIPIDS-TX, refers to medical interventions aimed at managing and correcting abnormal levels of lipids in the blood), glycated hemoglobin (HbA1c), glomerular filtration rate (GFR), glibenclamide (GB), metformin (MF), pioglitazone (PG), rosiglitazone (RG), acarbose (AB), and insuline (INS).

A total of 22 subject’s—diagnosed with T2D and T2D patients with DKD—averaging 57.9 ± 12.6, within a range of 36–84 years. Patient-specific variables included demographic, clinical observations and medical treatment. Furthermore, participant details, including continuous variables presented as mean ± SD, categorical variables expressed as n (%), and the associated *p*-values, are summarized in Table 2. The binary outcome for the classification model was defined as the presence or absence of DKD.

#### 2.1.2. Data Pre-Processing

Initially, the dataset contained a total number of 35 attributes; however, for the pre-processing step, the ID of each patient, neuropathy and retinopathy cases were removed because those input variables are not relevant for this study. Furthermore, in certain variables specifically, there are two missing observations in both variables DBP (9.0909%) and DBPU (9.0909%), while variable GFR has one missing observation (4.5454%). In this regard, a method employing the mean of the non-missing observations was employed to fill the missing values. According to Liu and Gopalakrishnan [30], the decision to utilize mean imputation is consistent with their findings, besides reinforcing the notion that imputation methods do not have a substantial impact on the results. As indicated by their research, there is no consistent superiority of one imputation method over another. In several cases, mean imputation has demonstrated its ability to yield results comparable to more complex algorithms.

### 2.2. Feature Selection Methodologies

#### 2.2.1. Boruta

The Boruta algorithm [31,32] is a relevant features selection wrapper method built around the RF classifier which provides important and non-important attributes from an information system. In the process, Boruta randomizes the input dataset (A_1_, A_2_, A_3_, …, A*_n_*) by adding copies of all the variables, named shadow attributes (S_1_, S_2_, S_3_, …, S*_n_*); then, the extended dataset is used to build an RF classifier in order to find the maximum Z-score among shadow attributes (MZSA). All the variables that scored lower than MZSA are treated as rejected, those with a higher MZSA are labeled as confirmed; finally, the remaining variables were marked as tentative.

#### 2.2.2. GALGO

GALGO employs genetic algorithms (GA) for selecting models with high fitness. This procedure starts from a random population of feature subsets known as chromosomes. Each chromosome is evaluated for its ability to classify the desirable outcome, obtaining a certain level of accuracy. The main idea is to replace the initial population with a new one including the variants of chromosomes with a higher classification accuracy; the progressive improvement of chromosome population is inspired by the process of natural selection (selection, mutation, and crossover). The proportion of the solution space increases with the chromosome populations in partiality isolated environments (niches); the chromosomes can migrate from one niche to another, ensuring the recombination of good solutions [29].

This process is carried out in four main steps:Setting up the analysis. The input and the outcome features are specified, as well as the parameters that define the GA environment (gene expression), the classification method, and the error estimation.Searching for relevant multivariate models. Starting from a random population of chromosomes, the GA method can find a diverse collection of good local solutions.Refinement and analysis of the population of selected chromosomes. The chromosomes selected from the GA are subjected to a backward selection strategy, aiming to obtain a model with a chromosome population that significantly contributes to the classification accuracy.Development of a representative statistical model. A single representative model is obtained; for this stage, a forward selection strategy was implemented based on the step-wise inclusion of the most prevalent genes in the chromosome population.

#### 2.2.3. Elastic Net

Zou and Hastie [33] proposed the elastic net, a shrinkage and selection method.

It is a convex combination of two different types of regularization methods: Ridge and LASSO regression ($L_{2},L_{1}$).

$L_{1}$, LASSO, a penalized least squares method imposing an $L_{1}$ penalty on the regression coefficients, does a continuous shrinkage and automatic variable selection.$L_{2}$, minimizes the residual sum of squares subject to a bound of the $L_{2}$-norm of the coefficients. Ridge regression achieves its better prediction performance through a bias-variance trade-off; however, this method cannot produce a parsimonious model due to it always keeping all the predictors.

The penalization method of the elastic net is defined by Equation (Equation 1):(1)$$\hat{\beta }\left(elastic-net\right)=argmin{∥y-X\beta ∥}_{2}^{2}+\lambda_{2}{∥\beta ∥}^{2}+\lambda_{1}{∥\beta ∥}_{1}$$
where *X* is the measurement matrix, and $\lambda_{2}$ and $\lambda_{1}$ are the shrinkage parameters.

The elastic net formula is shown in the Equation (Equation 2):(2)$$P_{\alpha }=\sum_{i=1}^{p}\left[\left(1-\alpha \right)\beta_{j}^{2}+\alpha \left|\beta_{j}\right|\right]$$
where α determines the specific type of regression; Ridge (alpha = 0) and LASSO (alpha = 1).

This union allows for learning a sparse model where few of the entries are non-zero, while still maintaining the regularization properties [34].

### 2.3. Classification Method

#### Random Forest

An ensemble method developed by Breiman [35]; RF is an ML algorithm that can perform classification and prediction tasks. This classifier provides a collection of decision trees (DT), in which each DT casts a unit vote for the most popular class; the final decision result of RF is obtained by the principle of majority, where the output of all the DT’s are aggregated to produce one final decision [36].

### 2.4. Evaluation Metrics

In this study, area under the curve (AUC), sensitivity (SN) and specificity (SP) were used to measure the effectiveness of the RF model. The formulas are as follows:(3)$$SN=\frac{TP}{TP+FN}$$
(4)$$SP=\frac{TN}{TN+FP}$$
where true positive (TP) represents the number of instances that are positive and correctly classified; true negative (TN) means the number of negative cases that are classified as negative; false positive (FP) is defined by negative instances that are incorrectly classified as positive, and false negative (FN), are the number of positive cases that are misclassified as negative. SN and SP analysis is commonly used for the evaluation of machine learning algorithms. SN has the capability of testing to correctly identify an individual as positive in disease (i.e., diabetic patients with DKD). The SP is the opposite of SN, this metric has the ability of a test to correctly classify an individual as disease-free (i.e., diabetic patients without DKD) [37].

Also, the receiver operating characteristic (ROC) curve and AUC are used to evaluate a classifier’s performance. The ROC curve, a two-dimensional measure, is a graph technique for visualizing the model efficiency, plotting the probability of correctly classifying (SN) against the incorrectly classifying (SP) examples. The AUC gives the probability that the classifier will rank a randomly chosen positive instance higher than a randomly chosen negative instance; when the AUC = 1.0 it indicates that the model can perfectly distinguish between high and low-risk patients, and when the AUC = 0.5, it is equivalent to random chance [38,39].

### 2.5. Cross Validation

A statistical method of evaluating and comparing machine learning algorithms is cross-validation (CV). One popular form of CV is a k-fold CV, where the dataset *D* is split randomly into *k* folds (subsets, $D_{1},D_{2},D_{3},\ldots ,D_{k}$), then the model is trained and tested *k* times, while the remaining k−1 folds are used for learning [40].

Figure 2, shows a representation of 10 fold CV.

### 2.6. Interpretability

SHapley Additive exPlanations (SHAP) [41] compute Shapley values, which quantify the contribution of each feature to the model’s predictions. The methodology involves evaluating the model’s output across various combinations of features, measuring the average change in prediction when a specific feature is included versus excluded. This difference, referred to as the Shapley value, represents the marginal contribution of a feature to the overall prediction. By providing a numerical assessment of the influence of each feature, SHAP enables a detailed interpretability of the model’s decision-making process [42,43].

The data were analyzed using R (version 4.0.3), a free statistical software. The following R packages were used for modeling: Boruta [31], MLeval [44], glmnet [45], GALGO [29], caret [46] and iml [47].

## 3. Results

### 3.1. Comparison to Established Feature Selection Methods

#### 3.1.1. Univariate and Multivariate Feature Selection

Univariate feature selection (UFS) examines the strength of the relationship between each variable and the response. In order to assess the affinity, several statistics approaches can be applied; however, in this case, the AUC value was calculated. In the context of FS, AUC is often used to evaluate the relevance of a feature by interpreting its values as the output of a classifier. Features with AUC values closer to 1 indicate high discriminative power, while those near 0.5 suggest no discriminatory ability [48]. This characteristic makes AUC particularly useful for identifying features that contribute meaningfully to the prediction task.

Table 3, presents 32 univariate models based on an RF approach, for those models that obtained an AUC value ≥ 0.60 a multivariate model was constructed.

For better comparison, firstly, the main approach was to include the 32 features and make an evaluation of the model. Therefore, aiming to develop a multivariate model established on the most significant features, seven selected variables were used as inputs for the RF algorithm.

Equation (Equation 5) represents the confirmed formula for the proposed multivariate model.
(5)OUTPUT∼GLU+UREA+CRE+TCHOLU+UTG+GFR+GB

#### 3.1.2. Boruta

Figure 3 shows the Boruta result plot. Blue boxplots correspond to minimal, average, and maximum Z-scores for each shadow attribute. Red, yellow, and green boxplots represent Z-scores of, respectively, the rejected, tentative and confirmed attributes.

Table 4, shows a summary of each attribute: mean, median, minimal, and maximal importance, the number of hits normalized to number of importance source runs, and the decision.

An important aspect to consider is that tentative attributes may be left without a decision; to fill the missing decisions, a simple comparison is performed using the median Z-score of the attributes with the median Z-score of the shadow attributes. The confirmed formula that will define a model based only on the confirmed attributes is presented in Equation (Equation 6):(6)OUTPUT∼CRE

#### 3.1.3. GALGO

Table 5, presents the general parameters used in GALGO.

To obtain a statistically significant model, the settings of this methodology were selected according to the literature [49,50]. Fifty-five feature RF models evolved throughout 300 generations. In each generation the chromosomes are selected, reproduced, crossed, and mutated, achieving an accurate and robust model. The fitness was defined as the accuracy of the model to classify the desirable outcome.

In Figure 4 it is possible to observe the most surviving features ordered by rank. From the plot, CRE and UREA show stable ranks while the remaining change the rank colors due to their instability.

The forward selection process is shown in Figure 5. The *y*-axis presents the classification accuracy, and the *x*-axis illustrates the genes ordered by their rank. In terms of accuracy, the model labeled as “1” containing the most three frequent genes was the best.

A robust gene backward elimination step was conducted, resulting in a representative model constructed with only three genes. The model obtained from GALGO with RF as a classifier is shown in Figure 6.

#### 3.1.4. Elastic Net

The Elastic net model used a combination of optimized values for Ridge (α = 0) and LASSO (α = 1) regressions. The values of the hyper-parameters α and λ were optimized by averaging five repetitions of 10-fold CV, in favor of minimizing the deviance error; thus, the optimal parameters of the elastic net were as follows: α = 0.555 and λ = 0.2.

In Figure 7, each curve corresponds to a variable. The x-axis shows λ for the log-lambda value. Furthermore, we indicate the variables going to zero as the penalty increases.

Table 6 summarizes the Elastic net performance, where 6 of 32 features were significant. Also, we show the shrunken coefficients for all the retained variables.

### 3.2. Classification Performance

According to the Fawcett criterion [38], the interpretation of the AUC values is as follows:Bad test = [0.5, 0.6)Regular test = [0.6, 0.75)Good test = [0.75, 0.9)Very good test = [0.9, 0.97)Excellent test = [0.97, 1)

These values were used to evaluate and interpret the efficiency of the models.

First, a supervised RF classification model was constructed; this model includes 32 features and employs a 10-fold CV strategy. Table 7, presents the results of the RF model, obtaining an AUC value of 0.60 indicating a bad performance. To estimate the probability of the disease the SN and SP were calculated.

Figure 8, shows the AUC obtained with the selected 32 features. The results established that the irrelevant variables degrade the performance of the model. This means that this model may not efficiently distinguish the outcome.

Subsequently, four RF classification models were constructed using a 10-fold CV strategy:Model 1. “feature obtained by Boruta [CRE]” + RFModel 2. “features obtained by GALGO [CRE, UREA and LIPIDS.TX]” + RFModel 3. “features obtained by Elastic net [AGE, UREA, CRE, LIPIDS.TX, GFR and GB]” + RFModel 4. “features obtained by UFS [GLU, UREA, CRE, TCHOLU, UTG, GFR, and GB]” + RF

The performance for each classification model was reported in Table 8. In the extreme case, Boruta, a tree-based method, selected only one attribute. However, the model showed a poor performance, achieving an AUC value of 0.61. On the contrary, GALGO, which uses a multivariate approach, obtained 0.80 of AUC representing a very good classification test with three features; in terms of SN (0.909) and SP (0.818), the model has the capability to diagnose individuals with and without the condition. On the other hand, Elastic net, a traditional method, is a combination of Ridge and LASSO penalty where the regression coefficients are shrinking near to zero; the model attained 0.75 of AUC with six variables, demonstrating an acceptable efficiency and a regular performance in terms of SN and SP. Finally, the seven characteristics were taken into account for a multivariate evaluation. The results showed regular discrimination (0.74 of AUC) but a lower ability of the model to distinguish the non-DKD subjects.

It should be noted that all methods have similarities. Despite the different methodologies of FS algorithms, it is important to note that CRE is a dominant attribute. Another biochemical indicator, the UREA, which is a strong variable, appears in the three FS method, on the contrary, GFR, LIPIDS.TX, and GB were consistent.

Figure 9, shows the AUC-ROC for each model that uses a different methodology of FS.

### 3.3. Model Interpretation

The GALGO feature selection method identified CRE, UREA, and LIPIDS.TX as the key attributes that best distinguish patients with DKD from those without. When these features were used in an RF classification model, it achieved high performance, with an AUC of 0.80. These findings demonstrate the effectiveness of GALGO in identifying clinically relevant predictors and the ability of RF to leverage these features for accurate classification. To provide greater interpretability of the model’s decision-making process, the SHAP algorithm was applied.

Figure 10 illustrates the feature importance scores computed using the SHAP methodology applied to the predictors of the Random Forest classifier. The feature importance is evaluated based on the cross-entropy loss, which reflects the influence of each variable on the predictive performance of the model. CRE has the highest importance score, and creatinine levels emerge as the most critical variable in predicting DKD; UREA demonstrates moderate importance reinforcing its clinical utility in assessing kidney function and its association with DKD progression, and LIPIDS.TX highlights the potential influence on DKD risk management and outcomes.

Figure 11 demonstrates how CRE levels influence the model’s predicted probability of DKD using SHAP combined with Partial Dependence Plots (PDPs).

The graph is divided into two panels: NO (NO-DKD) and YES (YES-DKD). In the NO panel, higher CRE levels correlate with a sharp decrease in the predicted probability of DKD, suggesting that elevated CRE levels reduce the likelihood of false-positive classifications among non-DKD patients. Conversely, in the YES panel, higher CRE levels correlate with an increased predicted probability of DKD, reaching a maximum at levels exceeding 1.0 mg/dL. In addition, the plot highlights a non-linear relationship between CRE and DKD predictions, emphasizing its asymmetric impact on the NO and YES groups.

Figure 12 illustrates the SHAP-based dependence plot for UREA levels. In the NO panel, the relationship between UREA levels and the predicted probability of DKD shows fluctuations, with a noticeable peak at approximately 25 mg/dL, followed by a decline and stabilization around 0.4 for higher UREA levels. This indicates that while UREA levels might momentarily increase the likelihood of misclassification, higher levels tend to reduce the probability of false positives among NO-DKD patients. In contrast, the YES panel shows an initial dip in predicted probability at UREA levels near 25 mg/dL, followed by a steady increase and stabilization around 0.6 for higher levels. This indicates that elevated UREA levels contribute positively to the classification of DKD, aligning with its established role as a maker of impaired renal function.

In the Figure 13, the NO panel shows the high predicted probability of DKD (~0.75) when LIPIDS.TX is inactive (0), which sharply declines to ~0.4 as LIPIDS.TX becomes active (1). Otherwise, the YES panel illustrates an increased probability of DKD, from ~0.3 when LIPIDS.TX is inactive to ~0.6 as LIPIDS.TX becomes active. These findings suggest a positive relationship between lipid therapy and the diagnosis of DKD.

## 4. Discussion and Conclusions

This work focused on comparing different feature selection (FS) methodologies to identify predictive risk factors of DKD using a dataset from Mexican T2D patients, both with and without DKD. Initially, the dataset comprised 47 features, but after a pre-processing stage to remove irrelevant features, it was narrowed down to 32 variables, as detailed in Table 2. The importance of identifying crucial features for disease diagnosis led to the adoption of feature dimension reduction methods such as UFS, Boruta, GALGO, and Elastic net, aimed at retaining significant information.

For the prediction of DKD or non-DKD status, a binary classifier was developed utilizing ML techniques. Among these, RF, an ensemble learning algorithm composed of multiple DTs, was selected for its extensive use in healthcare, particularly for developing prediction models. RF’s success is attributed to its ability to handle highly non-linear data, robustness to noise, and a simpler tuning process compared to other ensemble algorithms [51,52,53].

The attributes selected through FS methodologies were input into the RF classification algorithm, reducing the initial 32 candidate features to smaller subsets: 1 (Boruta), 3 (GALGO), 6 (Elastic net), and 7 (UFS). These subsets were then modeled using a 10-fold CV, demonstrating the effectiveness of this approach in achieving stable evaluations with a smaller dataset [54]. Table 8, demonstrates that the behavior of Elastic Net and the multivariate model, in terms of the variables selected, are pretty similar. Otherwise, Boruta and GALGO pick a fewer number of predictors.

The performance evaluation was carried out by quantitative metrics such as AUC, SN and SP. An ideal test has an AUC value of 1.0. However, an AUC < 0.5 is considered to have a reasonable discriminating ability, allowing for the description of the problem [55]. Then, according to the Fawcett criterion, the models were evaluated [38]. To begin, all the features were analyzed by the RF approach, the general model attained a 0.60 AUC value, but the performance terms of identifying DKD (SN) and non-DKD individuals (SP) was poor. Boruta + RF reached (0.61 AUC value) a similar performance with only one risk factor, but in comparison with the general model, the portion of negative instances was higher. On the other hand, Galgo + RF, had the most acceptable approach since it selects three variables, and records an AUC of 0.80, with the effectiveness of identifying cases and controls of the disease. Elastic net + RF can be an effective method, but it chooses an exhaustive set of attributes. It is important to note that compared with GALGO + RF, both employed the three same characteristics; nevertheless, this method selects a high number of variables and drops the precision of the model, obtaining an AUC of 0.75 which is a regular test, with equal SN and SP values (0.720). Finally, the UFS selects seven risk factors, those predictors formed a multivariate model which was evaluated in the same way as the other methodologies and achieved 0.74 of AUC, but in terms of SN and SP, the performance was deficient.

With the aim of comparing the outcomes derived from the RF algorithm against alternative machine learning approaches, we harnessed the three distinct features obtained by GALGO. Table 9 shows the metrics encompassing sensitivity, specificity and the AUC values for both RF and Support Vector Machine (SVM), elucidated across an array of cross-validation strategies.

The results exhibited that RF achieved superior performance, particularly underscored in the settings featuring k = 10 and Leave-One-Out Cross Validation (LOOCV); this is a validation method where each individual sample in the dataset is used once as a validation set while the remaining samples form the training set. The elevated AUC values accentuate RF’s adeptness in efficaciously discriminating between the classes, thereby increasing its utility in the precise classification of DKD and non-DKD patients. These findings serve to enhance the stature of RF as a valuable tool in the classification of DKD subjects, and the demonstrated robustness within cross-validation elucidates the reliability of RF across a spectrum of clinical scenarios.

The findings of the study highlight the importance of understanding different FS methods. Using appropriate modeling methods is a key component of this process. A generalization of the findings is that for clinical problems in a smaller dataset, GALGO, which is a genetic algorithm, obtains better results in order to classify patients with DKD.

Using GALGO and RF with only three features, the proposed model achieved impressive results with an AUC of 0.800, demonstrating its robustness. These outcomes are remarkable as they indicate that a parsimonious feature selection strategy can achieve a level of performance comparable to studies employing a more extensive set of variables (Table 10).

The nature-inspired methodologies, such as GALGO, extend their examination beyond linear associations between features and target outcomes. These techniques assess the collective potential of feature subsets rather than focusing solely on individual feature performance; while this approach may appear straightforward, it is essential to note that the combinatorial nature of feature subsets results in an exponential rise in the number of potential combinations, rendering their exhaustive evaluation computationally demanding within reasonable time frames. The application of a genetic algorithm contributes to the development of a resilient multivariate model [56].

The features obtained by this methodology are in accordance with several investigations mentioned above (Table 1). CRE and UREA are common biochemical indicators used to monitor the function of the kidney. CRE is an assured indicator of kidney function, which reflects the capacity of renal filtering; a high level of this biomarker is associated with poor clearance of CRE by the kidneys. Its sensitivity is poor in the early stages of renal impairment unless the damage is important enough to engage filtering [57,58]. UREA is an indicator of kidney damage and dysfunction and is helpful in giving an initial diagnosis [59].

From a molecular perspective, both CRE and UREA are closely tied to the pathophysiological mechanisms underpinning DKD. Persistent inflammation of the circulatory system and renal tissue serves as a critical pathophysiological basis for the development and progression of DKD. This inflammation can be triggered by metabolic, biochemical, and hemodynamic disturbances commonly observed in DKD [60]. Chronic hyperglycemia in diabetes patients induces oxidative stress and activates inflammatory pathways, leading to the production of pro-inflammatory cytokines such as IL-6 and TNF-α. These molecules contribute to tubular and glomerular injury, reducing renal filtration efficiency and increasing serum creatinine and urea levels. Persistent inflammation also promotes extracellular matrix deposition, leading to fibrosis, which exacerbates renal dysfunction [9,61].

With regard to LIPIDS.TX, the dataset of this study only mentions whether the patients are under medical lipids treatment. It should be noted that if the patient is undergoing the treatment, and efficient control is not achieved, there are deleterious effects of elevated cholesterol levels on renal injury and on the initiation and progression of DKD [62]. Likewise, Ayodele et al. [63] support that lipids play a role in the development and progression of glomerular injury. Dyslipidemia, a common feature in nephrotic syndrome and DKD, exacerbates renal damage through multiple mechanisms. The uptake of oxidized LDL by mesangial cells promotes glomerulosclerosis, while free fatty acids directly impair podocyte function, leading to cellular damage and loss. Additionally, dysregulated lipid metabolism, including altered cholesterol and fatty acid pathways, contributes to injury in glomerular and tubular cells. In DKD, mitochondrial dysfunction further aggravates these effects by disrupting lipid metabolism, inducing oxidative stress, and triggering inflammatory pathways, ultimately accelerating fibrosis and renal decline [64,65].

This research was carried out under the guidance of subject matter experts in diabetes and its associated conditions, guaranteeing that the methodology and findings align with clinical significance. This oversight ensured that the predictions and risk factors identified by the model were consistent with the current clinical understanding of DKD.

There are some limitations to this study. Firstly, data were only available from a limited set of patients with DKD. In total, data were available for only 22 patients (11 cases and 11 controls). Secondly, it may only be applicable to Mexican people; the cultural, genetic, and environmental factors unique to Mexico could influence the manifestation and progression of DKD, making it challenging to extrapolate the findings to other populations. Replicating the study in different ethnic groups and geographic regions would provide valuable insights into the generalizability of the results and help to identify potential population-specific factors contributing to the disease; Thirdly, it would be interesting to reproduce this methodology with the inclusion of other routine laboratory parameters that compromise the renal failure, and comorbidities, namely neuropathy and retinopathy data; including these additional variables could provide a more comprehensive understanding of factors influencing the development and progression of DKD. Furthermore, exploring the interactions between these attributes and identifying novel biomarkers predictive of disease outcomes could enhance the predictive accuracy and clinical utility of the model. It is crucial to acknowledge that the dataset lacks variables pertaining to physical activity and dietary habits, factors known to substantially impact the progression of diabetes and associated complications. This omission represents a limitation in the study, as incorporating these variables could have provided a more comprehensive perspective on the factors contributing to DKD.

Finally, in this context the attainment of highly effective models alone is insufficient. It is imperative to comprehend their functioning and their associations with input data. This is where explanatory techniques, such as SHAP and LIME (Local Interpretable Model-Agnostic Explanations), play a fundamental role. These techniques are pivotal for analyzing and comprehending how the model’s outcomes correlate with various input features, thereby providing clarity and transparency to the clinical decision-making process. For instance, the analyzed Figure 11, Figure 12 and Figure 13 demonstrate how key attributes such as CRE, UREA and LIPIDS.TX influence the model’s predictions in a differentiated manner. Specifically, the SHAP-based analysis underscores the non-linear relationships between CRE and UREA levels and the predicted probability of DKD, revealing distinct patterns for patients classified as non-DKD and DKD. These relationships validate the biological relevance of the variables and emphasize how the model utilizes these features to differentiate between DKD and non-DKD cases. Similarly, the impact of LIPIDS.TX underscores the importance of including therapeutic factors in clinical models, as it shows how treatment-related variables can influence predictions. The integration of explanatory algorithms not only facilitates a profound understanding of how model predictions are derived but also enables the identification of critical factors influencing medical decisions. This is essential for enhancing the quality of DM management and the treatment of its complications. Furthermore, the utilization of explanatory techniques empowers healthcare professionals to validate and refine the model’s performance. By analyzing the intricate relationships between input features and model outcomes, as demonstrated in the discussed Figure 11, Figure 12 and Figure 13, clinicians can identify potential biases or areas for improvement within the model. This iterative process of refinement not only enhances the reliability and accuracy of predictions but also fosters trust and confidence in the model’s capabilities among healthcare providers.

In summary, different FS methodologies were applied to identify predictors of DKD. Four main RF classification models are presented in this work (USF + RF, Boruta + RF, GALGO + RF and Elastic net + RF). Afterward, the performance measures were calculated. The results showed that the GALGO + RF model achieved 0.80 of AUC, obtaining promising results with only three risk factors: CRE, UREA, and LIPIDS.TX, permitting the effectiveness of classifying patients with DKD. There is a need to detect which risk factors participate in the development of DKD, diagnose the disease early, and determine the most effective medical treatments that are necessary to prevent the kidney damage progression. Based on the results, the model could be used to enable the development of a preliminary tool that can be useful to assist clinicians in the diagnosis of DKD and may improve their decision-making in the management of this disease. While the selected biomarkers are well-established in nephrology, their application within advanced FS methodologies such as GALGO outlines the potential of combining traditional markers in innovative ways to improve diagnostic performance. This approach not only validates the relevance of these biomarkers but also emphasizes their importance in creating highly sensitive and specific models for underrepresented populations, such as the Mexican cohort. The results evidence that leveraging ML techniques can uncover complex interactions between biomarkers, providing added clinical value beyond their independent contributions. As a future potential direction of research, the development of a hybrid model is proposed to integrate clinical data and imaging for improving the diagnosis and prediction of DKD progression. This model will combine comprehensive clinical information from patient records, including demographic data, medical history, laboratory test results, and prior treatments. Besides, medical imaging data such as ultrasounds will be collected to assess renal structure function. This future work has the potential to significantly improve the effectiveness of DKD diagnosis and management, leading to better patient outcomes and more personalized and efficient healthcare delivery.

## Figures and Tables

**Figure 1 biomedicines-12-02858-f001:**
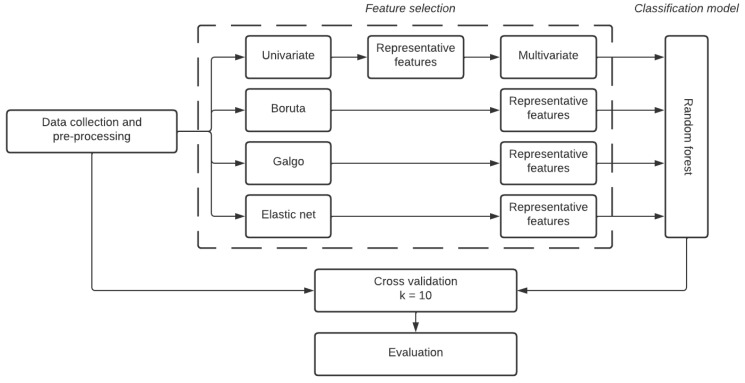
Architecture of the proposed classification system. (The value of *k* refers to the 10 folds used in the *k*-fold cross-validation process).

**Figure 2 biomedicines-12-02858-f002:**
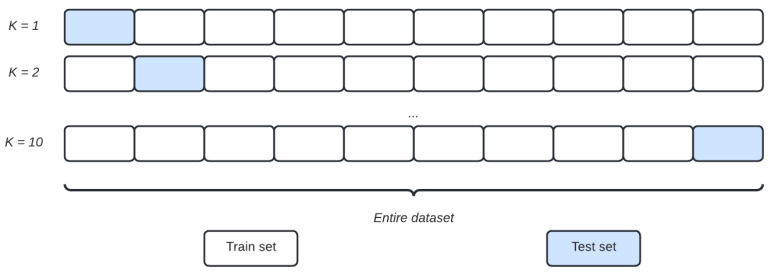
Procedure of 10-fold cross-validation. In this study, the entire dataset was divided into 10 subsets, nine of them were taken as the training data, and the rest were used as testing.

**Figure 3 biomedicines-12-02858-f003:**
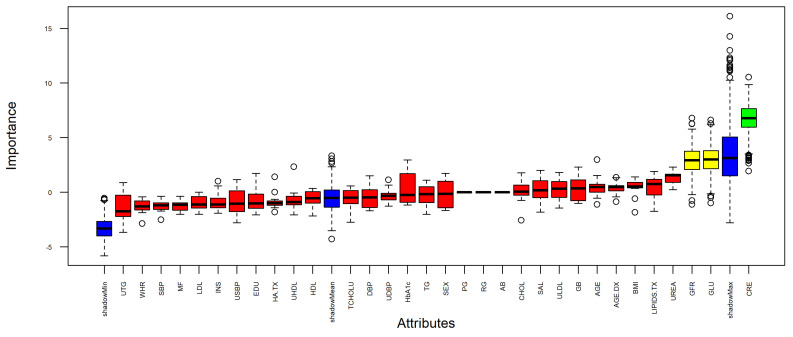
Boxplots of attributes based on importance values. (The horizontal line within each box represents the median value, and the white circles indicate outliers).

**Figure 4 biomedicines-12-02858-f004:**
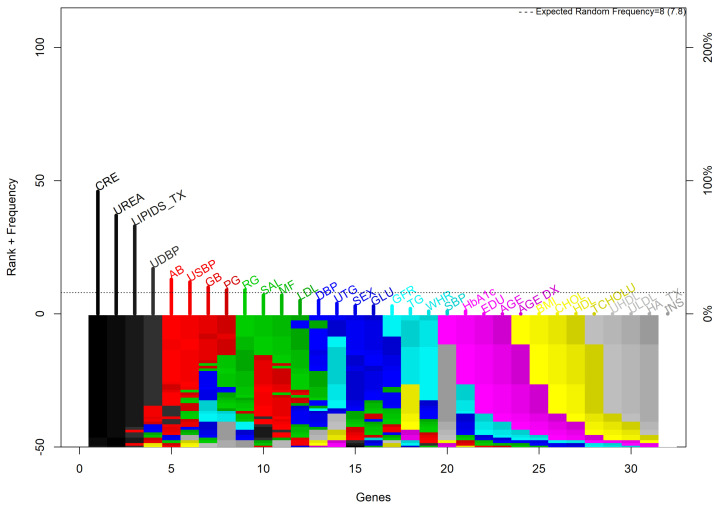
Gene rank stability.

**Figure 5 biomedicines-12-02858-f005:**
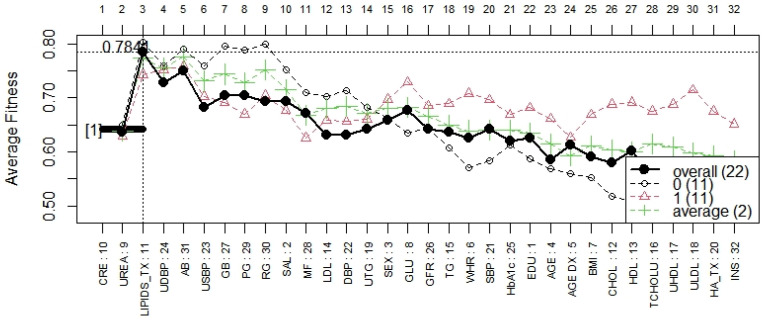
Forward selection models.

**Figure 6 biomedicines-12-02858-f006:**
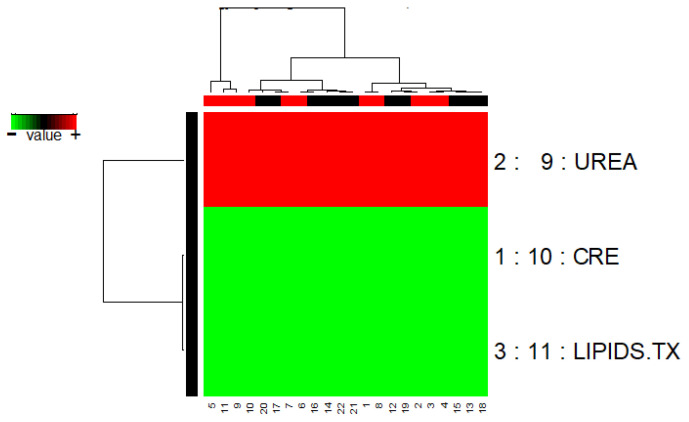
The model obtained from GALGO with RF as a classifier.

**Figure 7 biomedicines-12-02858-f007:**
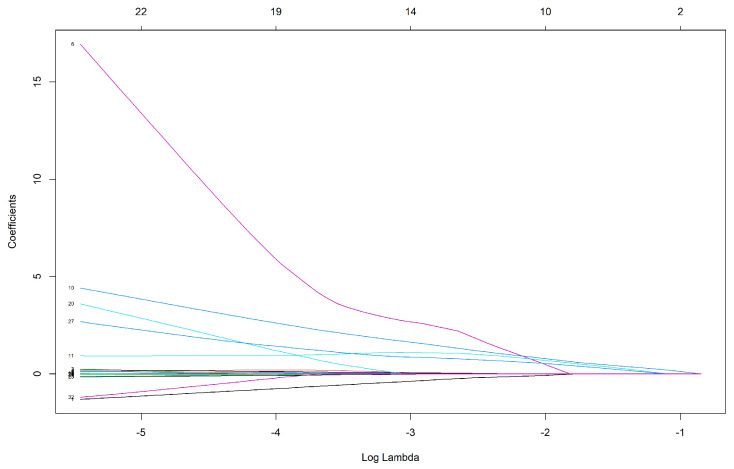
Elastic net regularized coefficients are shown as a function of log. Each line represents the coefficient path of a feature as the regularization penalty changes. The colors correspond to different features in the model.

**Figure 8 biomedicines-12-02858-f008:**
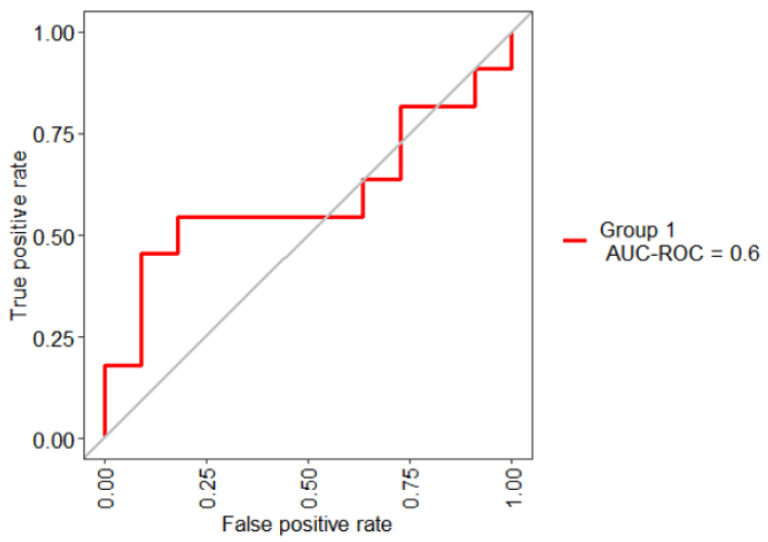
ROC curve for the DKD diagnosis by using the entire dataset.

**Figure 9 biomedicines-12-02858-f009:**
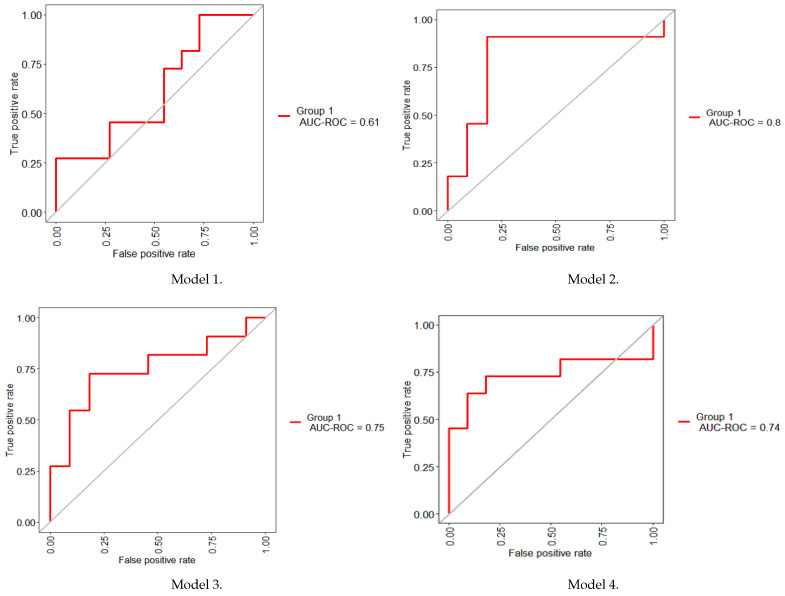
ROC curves of the four RF classification models.

**Figure 10 biomedicines-12-02858-f010:**
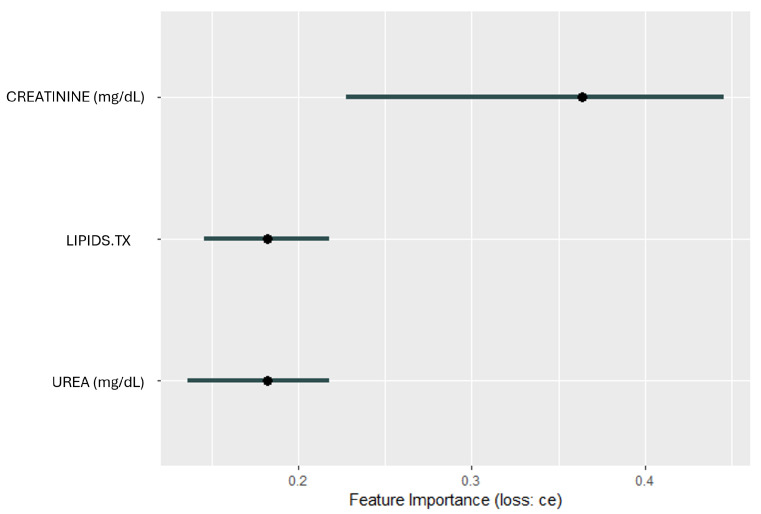
Feature importance derived from Shapley values for the top predictors. The predictors include creatinine (CRE), lipids treatment (LIPIDS.TX) and urea (UREA). The horizontal bars indicate the contribution of each feature to the model’s predictions, measured by Shapley values, with CRE showing the highest impact.

**Figure 11 biomedicines-12-02858-f011:**
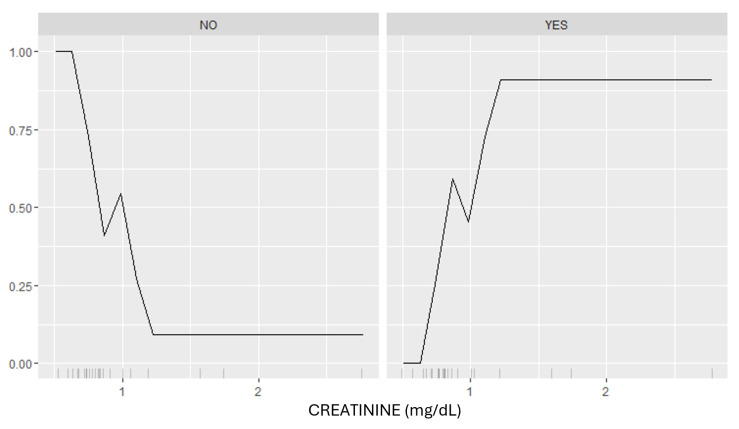
Dependence plot—effect of CRE levels on the predicted probability of DKD. The **left panel** (‘NO’) represents NO-DKD patients, where higher CRE levels correspond to a lower probability of misclassification. The **right panel** (‘YES’) represents YES-DKD patients, where increasing CRE levels are positively associated with the likelihood of DKD prediction.

**Figure 12 biomedicines-12-02858-f012:**
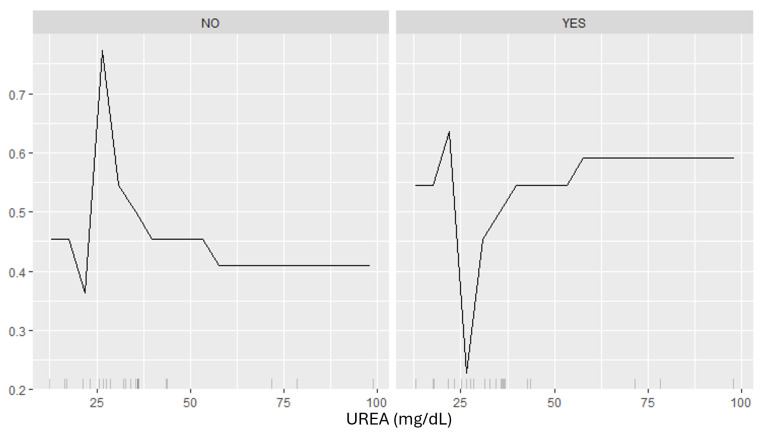
Dependence plot—effect of UREA levels on the predicted probability of DKD. The **left panel** (‘NO’) represents NO-DKD patients, showing fluctuating probabilities with a peak around 25 mg/dL, followed by stabilization at lower probabilities as urea levels increase. The **right panel** (‘YES’) represents YES-DKD patients, where urea levels initially cause a dip in predicted probability but then steadily increase, stabilizing at higher probabilities beyond 50 mg/dL.

**Figure 13 biomedicines-12-02858-f013:**
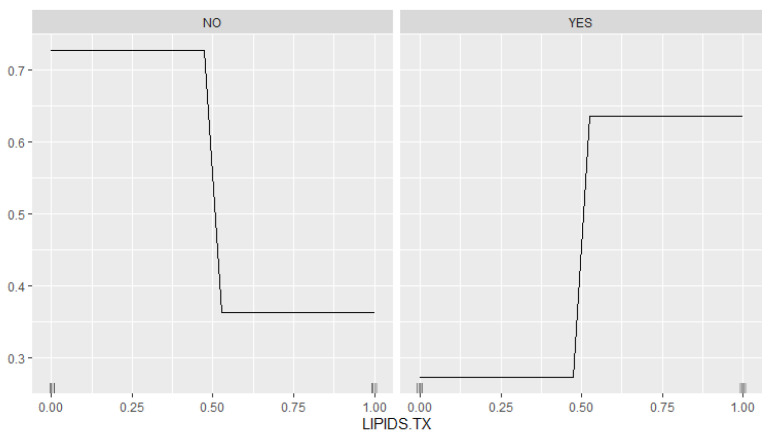
Dependence plot—effect of LIPIDS.TX on the predicted probability of DKD. The **left panel** (‘NO’) represents NO-DKD patients, showing a high predicted probability of misclassification for those without lipid treatment, which sharply declines beyond a certain threshold of treatment presence. The **right panel** (‘YES’) represents YES-DKD patients, where the predicted probability of DKD increases with the presence of lipid treatment, stabilizing at higher probabilities as treatment is applied.

**Table 1 biomedicines-12-02858-t001:** Summary of reviewed studies addressing DKD prediction in T2D patients.

Author’s and Year	Feature Selection Method	Risk Factors	Classification Algorithm
Rodriguez-Romero et al. 2019 [20]	InfoGain	Age Cholesterol Triglycerides Low-density lipoprotein Urinary albumin excretion Glomerular filtrarion rate	1R J48 RF SL SMO NB
Jiang et al. 2020 [21]	LASSO	DR HbA1c Gender Anemia Hemateuria DM duration Blood pressure Urinary protein excretion Estimated glomerular filtrarion rate	LR
Shi et al. 2020 [22]	LASSO LR	HbA1c Disease course Body mass index Total triglycerides Blood urea nitrogen Systolic blood pressure Postprandial blood glucose	LR
Xi et al. 2021 [23]	LASSO LR	Age Gender Hypertension Medicine use Duration of DM Body mass index Serum creatinine level Blood urea nitrogen level Neutrophil to lympocyte ratio Red blood cell distribution width	LR
Maniruzzaman et al. 2021 [24]	PCA	Age HbA1c Triglycerides DM duration Body mass index Fasting blood sugar Low density lipoprotein High density lipoprotein Systolic blood pressure Diastolic blood pressure	LDA SVM LR K-NN NB ANN
Yang and Jiang 2022 [25]	Univariate AIC	HbA1c Triglycerides Hypertension Serum creatinine Body mass index Blood urea nitrogen Diabetic peripheral neuropathy	LR

Abbreviations: Diabetic Kidney Disease (DKD); Type 2 Diabetes (T2D); Information Gain (InfoGain); Least Absolute Shrinkage and Selection Operator (LASSO); Diabetic Retinopathy (DR); Glycated Hemoglobin (HbA1c); Diabetes Mellitus (DM); Logistic Regression (LR); Principal Component Analysis (PCA); Linear Discriminant Analysis (LDA); Support Vector Machine (SVM); K-Nearest Neighbors (K-NN); Naive Bayes (NB); Artificial Neural Network (ANN); Akaike Information Criterion (AIC).

**Table 2 biomedicines-12-02858-t002:** Summary statistics of the entire dataset.

Predictors (n = 32)	Cases (n = 11)	Controls (n = 11)	*p*-Value
**Demographic Characteristics**			
EDU			
1—Elementary school 2—Secondary school 3—Technical school 4—High school 5—Professional 6—Postgraduate	2 (18.18%) 3 (27.27%) 2 (18.18%) - 4 (36.36%) -	1 (9.09%) 2 (18.18%) 2 (18.18%) 3 (27.27%) 3 (27.27%) -	0.559
SAL			
1—Less than 2000.00 MXN (~98.16 USD) 2—Between 2000.00 and 5000.00 MXN (~98.16–245.39 USD) 3—More than 5000.00 MXN (~245.39 USD)	3 (27.27%) 3 (27.27%) 5 (45.45%)	3 (27.27%) 2 (18.18%) 6 (54.54%)	0.409
SEX			
0—Male 1—Female	6 (54.54%) 5 (45.45%)	6 (54.54%) 5 (45.45%)	1
AGE, years	62.909 (±11.597)	52.909 (±12.045)	**0.0816**
AGE DX, years	45.636 (±6.561)	45.272 (±11.199)	0.922
**Clinical observations**			
WHR, cm/cm	0.963 (±0.092)	0.914 (±0.094)	0.233
BMI kg/m^2^	30.486 (±3.685)	29.225 (±7.205)	0.591
SBP, mmHg	125.909 (±15.623)	1223.636 (±15.666)	0.722
DBP, mmHg	82.727 (±9.045)	79.545 (±8.790)	0.395
USBP, mmHg	122.272 (±12.915)	120.909 (±12.21)	0.79
UDBP, mmHg	80.909 (±8.312)	78.181 (±7.507)	0.413
GLU, mg/dL	127.363 (±43.845)	167.363 (±78.888)	0.159
UREA, mg/dL	45.818 (±25.732)	28 (±7.835)	0.1082
CRE, mg/dL	1.196 (±0.622)	0.733 (±0.136)	**0.074**
CHOL, mg/dL	222.518 (±74.48)	206.663 (±58.252)	0.475
HDL, mg/dL	37.881 (±10.939)	40.136 (±14.742)	0.672
LDL, mg/dL	161.045 (±67.839)	136.055 (±37.055)	0.294
TG, mg/dL	247.427 (±162.302)	310.536 (±274.891)	0.503
TCHOLU, mg/dL	193.909 (±69.288)	193 (±41.916)	0.969
UHDL, mg/dL	41.639 (±9.982)	41.0909 (±14.117)	0.913
ULDL, mg/dL	130.181 (±60.237)	123.545 (±22.979)	0.722
UTG, mg/dL	211.090 (±151.341)	302 (±260.799)	0.325
HBA1C, mmol/L	6.994 (±2.527)	8.209 (±2.667)	0.28
GFR	76.807 (±28.853)	111.454 (±44.331)	**0.0583**
**Medical treatment**			
LIPIDS-TX			
0—No 1—Yes	4 (36.36%) 7 (63.63%)	8 (72.72%) 3 (27.27%)	**0.095**
HA-TX			
0—No 1—Yes	7 (63.63%) 4 (36.36%)	8 (72.72%) 3 (27.27%)	0.48
GB			
0—No 1—Yes	6 (54.54%) 5 (45.45%)	10 (90.90%) 1 (9.09%)	**0.08**
MF			
0—No 1—Yes	3 (27.27%) 8 (72.72%)	3 (27.27%) 8 (72.72%)	1
PG			
0—No 1—Yes	11 (100%) -	11 (100%) -	-
RG			
0—No 1—Yes	11 (100%) -	11 (100%) -	-
AB			
0—No 1—Yes	11 (100%) -	11 (100%) -	-
INS			
0—No 1—Yes	8 (72.72%) 3 (27.27%)	6 (54.54%) 5 (45.45%)	0.379

**Table 3 biomedicines-12-02858-t003:** Univariate analysis: AUC values of each feature.

Feature	AUC Value	Feature	AUC Value
EDU	0.29	LDL	0.29
SAL	0.55	TG	0.50
SEX	0.26	TCHOLU	0.64
AGE	0.53	UHDL	0.58
AGE DX	0.47	ULDL	0.55
WHR	0.25	UTG	0.67
BMI	0.57	HBA1C	0.46
SBP	0.13	GFR	0.68
DBP	0.28	GB	0.60
USBP	0.40	MF	0.16
UDBP	0.41	PG	-
GLU	0.67	RG	-
UREA	0.60	AB	-
CRE	0.61	INS	0.45
CHOL	0.53	LIPIDS.TX	0.48
HDL	0.44	HA TX	0.24

**Table 4 biomedicines-12-02858-t004:** Attributes importance stats.

Attribute	meanImp	medianImp	minImp	maxImp	normHits	Decision
EDU	−0.68	−1.02	−2.07	1.73	0.00	Rejected
SAL	0.23	0.18	−1.81	2.00	0.00	Rejected
SEX	−0.44	−0.725	−1.69	1.41	0.00	Rejected
AGE	0.81	1.04	−0.99	2.05	0.00	Rejected
AGE DX	0.01	0.21	−1.96	1.93	0.00	Rejected
WHR	−1.10	−1.36	−2.14	0.80	0.00	Rejected
BMI	0.05	0.09	−1.01	1.20	0.00	Rejected
GLU	3.01	2.98	−0.65	6.61	0.46	Tentative
UREA	0.92	0.96	−1.07	2.34	0.00	Rejected
CRE	6.69	6.83	2.16	9.81	0.85	Confirmed
LIPIDS-TX	0.12	0.12	−2.07	2.23	0.00	Rejected
CHOL	−0.02	0.37	−2.90	1.16	0.00	Rejected
HDL	−0.43	−0.38	−2.20	1.56	0.00	Rejected
LDL	−1.00	−0.96	−2.13	0.07	0.00	Rejected
TG	−0.72	−0.59	−2.60	1.10	0.00	Rejected
TCHOLU	−0.54	−0.98	−2.66	1.76	0.00	Rejected
UHDL	−1.08	−1.06	−3.28	0.42	0.00	Rejected
ULDL	0.01	−0.01	−2.70	2.11	0.00	Rejected
UTG	−1.08	−0.98	−2.32	−0.16	0.00	Rejected
HA-TX	−0.56	−0.95	−1.62	1.38	0.00	Rejected
SBP	−0.81	−0.74	−2.47	1.41	0.00	Rejected
DBP	−0.81	−0.89	−2.08	0.63	0.00	Rejected
USBP	−1.03	−1.25	−2.13	0.48	0.00	Rejected
UDBP	−0.51	−0.46	−1.66	0.75	0.00	Rejected
HBA1C	0.08	0.15	−1.46	2.21	0.00	Rejected
GFR	2.95	3.00	−0.75	7.23	0.48	Tentative
GB	0.13	−0.22	−1.16	2.05	0.00	Rejected
MF	−0.92	−1.00	−2.08	1.00	0.00	Rejected
PG	0.00	0.00	0.00	0.00	0.00	Rejected
RG	0.00	0.00	0.00	0.00	0.00	Rejected
AB	0.00	0.00	0.00	0.00	0.00	Rejected
INS	−0.75	−0.85	−1.72	0.87	0.00	Rejected

**Table 5 biomedicines-12-02858-t005:** GALGO input parameters.

Parameter	Description	Value
Classification method	Method used for classification	RF
Chromosome size	Number of variables include in each model	5
Max solutions	A collection of solutions desired	50
Max generations	Number of generations that GA can evolve	300
Goal fitness	Desired fitness value	1.0

**Table 6 biomedicines-12-02858-t006:** Selected features of elastic net analysis.

Features	Coefficients
AGE	0.010
UREA	0.001
CRE	0.494
LIPIDS-TX	0.406
GFR	−0.004
GB	0.310

**Table 7 biomedicines-12-02858-t007:** Evaluation metrics for the model with 32 features.

AUC	SN	SP
0.60	0.545	0.455

**Table 8 biomedicines-12-02858-t008:** Performance of models together with the number of variables each feature selection method.

Feature Selection Method	Features Number	Features	AUC	SN	SP
Boruta	1	CRE	0.61	0.455	0.636
Galgo	3	CRE UREA LIPIDS.TX	0.80	0.909	0.818
Elastic net	6	AGE UREA CRE LIPIDS.TX GFR GB	0.75	0.720	0.720
Multivariate	7	GLU UREA CRE TCHOLU UTG GFR GB	0.74	0.636	0.545

**Table 9 biomedicines-12-02858-t009:** Performance comparison among different cross-validation strategies and classification models.

**LOOCV + (3 FEAT [GALGO]) RF**			**LOOCV + (3 FEAT [GALGO]) SVM**		
**AUC**	**SN**	**SP**	**AUC**	**SN**	**SP**
0.790	0.818	0.818	0.760	0.727	0.818
**k = 10 CV + (3 FEAT [GALGO]) RF**			**k = 10 CV + (3 FEAT [GALGO]) SVM**		
**AUC**	**SN**	**SP**	**AUC**	**SN**	**SP**
0.800	0.909	0.818	0.750	0.727	0.727

**Table 10 biomedicines-12-02858-t010:** Comparative overview of our study and previous investigations.

Author’s and Year	Feature Selection Method	# of Features	Classification Algorithm	Evaluation Metrics
Rodriguez-Romero et al. 2019 [20]	InfoGain	6	1R J48 RF SL SMO NB	SN: 0.871; SP: 0.979; ACC: 0.871 SN: 0.887; SP: 0.249; ACC: 0.887 SN: 0.887; SP: 0.999; ACC: 0.887 SN: 0.899; SP: 0.183; ACC: 0.899 SN: 0.893; SP: 0.159; ACC: 0.893 SN: 0.804; SP: 0.355; ACC: 0.803
Jiang et al. 2020 [21]	LASSO	9	LR	C-INDEX: 0.934
Shi et al. 2020 [22]	LASSO LR	10	LR	AUC: 0.807; C-INDEX: 0.807
Xi et al. 2021 [23]	LASSO LR	10	LR	AUC: 0.813; C-INDEX: 0.819
Maniruzzaman et al. 2021 [24]	PCA	10	LDA SVM-RBF LR KNN NB ANN	AUC: 0.880; SN: 0.867; SP: 0.852 AUC: 0.910; SN: 0.867; SP: 0.863 AUC: 0.890; SN: 0.800; SP: 0.823 AUC: 0.890; SN: 0.833; SP: 0.849 AUC: 0.900; SN: 0.717; SP: 0.904 AUC: 0.900; SN: 0.800; SP: 0.823
Yang and Jiang 2022 [25]	Univariate AIC	7	LR	AUC: 0.758; C-INDEX: 0.758
**Proposed**	GALGO	3	RF	AUC: 0.800; SN: 0.909; SP: 0.818

## Data Availability

No new data data were created. Data sharing is not applicable to this article.

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
