# Peer review of "Evaluating Feature Selection Methods for Accurate Diagnosis of Diabetic Kidney Disease"

_biomedicines, 2024, doi:10.3390/biomedicines12122858_

Round 1
Reviewer 1 Report
Comments and Suggestions for Authors
I am really grateful to review this manuscript. In my opinion, this manuscript can be published once some revision is done successfully. I made one suggestion and I would like to ask your kind understanding. The application of traditional statistical approaches in diabetic kidney disease (DKD) shows limited success given that there might exist a variety of potential predictors unknown for this comorbid condition. Machine learning is known for its great performance with feature selection but little literature is available in this direction. For this reason, this study attempted to evaluate the usefulness of machine learning as predictive artificial intelligence regarding DKD in Mexico. This study used numeric data from 22 participants enrolled in the Mexican Social Security Institute (IMSS), applied the random forest with four feature selection approaches and achieved the area under the curve of 80% from Galgo, followed by the elastic net (75%), the multivariate composition (74%) and Boruta (61%). This study presented a systematic comparison of previous studies and variable importance outcomes from the random forest with Galgo as well. I would argue that this is a good achievement.
However, it can be noted that the random forest Shapley Additive Explanations (SHAP) dependence plot is very effective to identify the strength and direction of association between high fertility rates and its major predictor. In this context, I would like to ask the authors to derive the random forest SHAP dependence plot. For reference, I would like to introduce brief introductions here. The SHAP value of a predictor for a participant measures the difference between what the random forest predicts for the probability of DKD with and without the predictor. For example, let’s assume that the SHAP values of lipid treatment for DKD have the range of (0.2, 0.4). Here, some participants have SHAP values as low as 0.2, and other participants have SHAP values as high as 0.4. The inclusion of the predictor (lipid treatment) into the random forest will increase the probability of the dependent variable (DKD) by the range of 0.2 and 0.4. Here, the SHAP dependence plot is expected to (1) show there exists a positive association between the predictor (lipid treatment) and the dependent variable (DKD) and (2) test whether there exists a non-linear association.
Comments on the Quality of English LanguageThe English could be improved to more clearly express the research.
Author Response
Reviewer 1
I am really grateful to review this manuscript. In my opinion, this manuscript can be
published once some revision is done successfully. I made one suggestion and I would like
to ask your kind understanding. The application of traditional statistical approaches in
diabetic kidney disease (DKD) shows limited success given that there might exist a variety of
potential predictors unknown for this comorbid condition. Machine learning is known for its
great performance with feature selection but little literature is available in this direction. For
this reason, this study attempted to evaluate the usefulness of machine learning as
predictive artificial intelligence regarding DKD in Mexico. This study used numeric data from
22 participants enrolled in the Mexican Social Security Institute (IMSS), applied the random
forest with four feature selection approaches and achieved the area under the curve of 80%
from Galgo, followed by the elastic net (75%), the multivariate composition (74%) and Boruta
(61%). This study presented a systematic comparison of previous studies and variable
importance outcomes from the random forest with Galgo as well. I would argue that this is a
good achievement.
However, it can be noted that the random forest Shapley Additive Explanations (SHAP)
dependence plot is very effective to identify the strength and direction of association
between high fertility rates and its major predictor. In this context, I would like to ask the
authors to derive the random forest SHAP dependence plot. For reference, I would like to
introduce brief introductions here. The SHAP value of a predictor for a participant measures
the difference between what the random forest predicts for the probability of DKD with and
without the predictor. For example, let’s assume that the SHAP values of lipid treatment for
DKD have the range of (0.2, 0.4). Here, some participants have SHAP values as low as 0.2,
and other participants have SHAP values as high as 0.4. The inclusion of the predictor (lipid
treatment) into the random forest will increase the probability of the dependent variable
(DKD) by the range of 0.2 and 0.4. Here, the SHAP dependence plot is expected to (1) show
there exists a positive association between the predictor (lipid treatment) and the dependent
variable (DKD) and (2) test whether there exists a non-linear association.
Authors’ answer: Thank you for your valuable suggestion regarding the inclusion of a
SHAP dependence plot and its utility in understanding the association between predictors
and the dependent variable. We have addressed your comments thoroughly by making the
following additions to the manuscript:
● Added and interpretability section: 2.6 Interpretability, this section provides and
overview of the SHAP methodology.
● Included a model interpretation section: 3.3 Model interpretation, this section
present the three SHAP-based dependence plots, and each plot is analyzed in detail
to explain: (1) the strength and direction of association between the predictors and
the probability of DKD; (2) Any observed non-linear relationships, as highlighted in
the case of LIPIDS.TX where SHAP values demonstrate a clear positive association.
● The discussion section has been expanded to incorporate the insights derived from
the SHAP dependence plots (581-602)

Reviewer 2 Report
Comments and Suggestions for Authors
I think the conclusions from the current is stdy is already know for nephrologist, and I did not find any novel information from manuscirpt.
Serum creatinine and urine protein have been widely used as biomarker for diabetic nephropathy, I recommend that authors choose novel biomarker, such as KIM-1, NGAL or miRNA.
Author Response
Reviewer 2
I think the conclusions from the current is stdy is already know for nephrologist, and I did not
find any novel information from manuscirpt.
Serum creatinine and urine protein have been widely used as biomarker for diabetic
nephropathy, I recommend that authors choose novel biomarker, such as KIM-1, NGAL or
miRNA.
Authors’ answer: Thank you for your feedback and recommendations regarding the
manuscript. We appreciate your perspective and acknowledge the importance of novel
biomarkers in advancing the field of DKD research.
We agree that CRE and UREA are widely recognized as traditional biomarkers for DKD.
However, our study aims to demonstrate the utility of advanced machine learning
approaches in extracting meaningful patterns from these well-established markers.
Specifically, the inclusion of a feature selection like GALGO allowed us to identify
combinations of markers - beyond their independent contributions - that maximize model
performance. By taking advantage of these combinations, the model uncovers complex
relationships between features, which translates into greater sensitivity and specificity,
especially in the context of the Mexican population.
The suggestion to incorporate emerging biomarkers such as KIM-1, NGAL or miRNA. While
these biomarkers undoubtedly hold significant potential, our dataset was limited to routinely
available clinical markers, which reflect real-world clinical practice in our population.
Nevertheless, the use of GALGO and SHAP in this study highlights the potential of
combining common biomarkers in novel ways to achieve robust predictions. Future research
can extend this work by incorporating these innovative biomarkers, as suggested, to further
enhance the model’s predictive power and explore interactions between traditional and
emerging biomarkers.
To address your feedback further, we have added a paragraph to the conclusions,
emphasizing that while selected biomarkers are well-known, their use in combination with
advanced feature selection algorithms like GALGO allows the identification of interactions
that enhance diagnostic performance. This is particularly important in underrepresented
populations, where routine biomarkers remain the mainstay of diagnosis. (lines 613-620)

Reviewer 3 Report
Comments and Suggestions for Authors
1. The term “diabetes mellitus” is now recognized solely as “diabetes” by the International Diabetes Federation (IDF).
2. What do you mean by the term "lipids treatment" (line 31)?
3. It would be advisable to convert the DKD 2018 expenses (line 57) and the patient income from Table 2 into US dollars. Both expressions could be retained, not forgetting the currency abbreviations: MXN and USD.
4. You need to cite and add current references for the introduction and discussion.
5. Review any abbreviations you define before you use them for the first time (for example, LASSO on line 107). Also, keep the same format (for example, LASSO on line 121).
6. It is convenient to add the list of abbreviations in table 1.
7. It would be useful to indicate what the value of k in Figure 1 means.
8. The age of control patients and patients with chronic kidney disease should not be significantly different. This is a variable that can be controlled with inclusion and exclusion criteria.
9. Justify why the AUC parameter was used to evaluate the affinity in the UFS.
10. Albuminuria is a very important parameter as a diagnostic criterion for DKD. Why was it not included in the study among the parameters to be evaluated? Please explain this absence. Although this model aims to identify DKD in early stages, it should not be forgotten that albuminuria is a determining clinical parameter that cannot be ignored (despite appearing in the later stages of the disease).
11. What does LOOCV mean on line 433?
12. Please indicate what the scope of this model would be. Please also indicate whether the method was validated by a health specialist in this disease to verify the correlation of the predictions with clinical values.
13. A more detailed description of the molecular pathophysiology of diabetic kidney disease would further facilitate the relationship with the evaluated parameters.
14. It would be advisable to include the methodology for selection (inclusion and exclusion parameters).
15. The number of patients evaluated is too small to reach robust conclusions, and this should be explored further in the limitations of the study. In addition, the duration of diabetes, physical activity and diet would be variables of utmost importance to consider or, where appropriate, indicate in the limitations of the study..
16. Verify that the reference list complies with the guidelines for authors.
Author Response
1. The term “diabetes mellitus” is now recognized solely as “diabetes” by the
International Diabetes Federation (IDF).
Authors’ answer: Thank you very much for your observation. We acknowledge that the
International Diabetes Federation (IDF) now recognizes the term “diabetes mellitus” as
simply “diabetes”. We have updated the manuscript.
2. What do you mean by the term "lipids treatment" (line 31)?
Authors’ answer: Thank you for your question. By lipids treatment, we refer to whether or
not the patients received treatment aimed at correcting lipids levels.
3. It would be advisable to convert the DKD 2018 expenses (line 57) and the patient
income from Table 2 into US dollars. Both expressions could be retained, not
forgetting the currency abbreviations: MXN and USD.
Authors’ answer: Thank you for your suggestion. We have made the requested changes.
Both expressions, in Mexican pesos (MXN) and USD.
4. You need to cite and add current references for the introduction and discussion.
Authors’ answer: Thank you for your observation. We have added several current
references to both the introduction and discussion sections to support the statements made
and provide updated insights into the study
5. Review any abbreviations you define before you use them for the first time (for
example, LASSO on line 107). Also, keep the same format (for example, LASSO on
line 121).
Authors’ answer: Thank you for your observation. We have reviewed and corrected all
abbreviations to ensure they are defined before their first use and consistently formatted
throughout the manuscript.
6. It is convenient to add the list of abbreviations in table 1.
Authors’ answer: Thank you for the suggestion. The list of abbreviations has been added to
Table 1 as requested.
7. It would be useful to indicate what the value of k in Figure 1 means.
Authors’ answer: Thank you for your observation. We have added a legend to the caption
of Figure 1 clarifying that the value of “k” represents the 10 folds used in the k-fold
cross-validation process.
8. The age of control patients and patients with chronic kidney disease should not be
significantly different. This is a variable that can be controlled with inclusion and
exclusion criteria.
Authors’ answer: Thank you for your observation. We have added a table outlining the
inclusion criteria for cases and controls, along with the exclusion criteria. The individuals
included in the study were those recruited as part of the original project. (lines 173-197)
9. Justify why the AUC parameter was used to evaluate the affinity in the UFS.
Authors’ answer: Thank you very much for your comments. We have addressed your
observation by including a detailed justification, regarding the use of the AUC parameter to
evaluate the affinity in the UFS process. (Section 3.1.1, lines: 329-334)
10. Albuminuria is a very important parameter as a diagnostic criterion for DKD. Why
was it not included in the study among the parameters to be evaluated? Please
explain this absence. Although this model aims to identify DKD in early stages, it
should not be forgotten that albuminuria is a determining clinical parameter that
cannot be ignored (despite appearing in the later stages of the disease).
Authors’ answer: Thank you very much for your comments. The study did not include
albuminuria as a parameter due to limitations in the available dataset, which does not
contain clinical variables directly related to kidney function.
11. What does LOOCV mean on line 433?
Authors’ answer: Thank you very much for your observation. We have clarified the
meaning of LOOCV in the manuscript. (Lines 504-506)
12. Please indicate what the scope of this model would be. Please also indicate
whether the method was validated by a health specialist in this disease to verify the
correlation of the predictions with clinical values.
Authors’ answer: Thank you very much for your comment. The scope of this model is to
provide a reliable tool for the early identification of diabetic kidney disease in patients with
T2D, focusing on minimizing the clinical and economic burden associated with delayed
diagnosis. Regarding the validation of the method, one of the authors is affiliated with a
leading research center focused on diabetes and its complications. This expertise has
ensured that the model’s predictions are thoroughly evaluated and correlated with clinical
values, enhancing its reliability and relevance.
13. A more detailed description of the molecular pathophysiology of diabetic kidney
disease would further facilitate the relationship with the evaluated parameters.
Authors’ answer: Thank you for your suggestion. We have added a more detailed
description of the molecular pathophysiology of DKD to the manuscript. (lines 537-546,
552-559)
14. It would be advisable to include the methodology for selection (inclusion and
exclusion parameters).
Authors’ answer: Thank you for your observation. We have included the methodology for
selection, specifying the inclusion and exclusion criteria. (lines 173-197)
15. The number of patients evaluated is too small to reach robust conclusions, and
this should be explored further in the limitations of the study. In addition, the duration
of diabetes, physical activity and diet would be variables of utmost importance to
consider or, where appropriate, indicate the limitations of the study.
Authors’ answer: Thank you for your observation. We have addressed this comment by
adding lines in the discussion and conclusion section to acknowledge the absence of
variables such as physical activity and diet in the dataset. (lines: 537-577)
16. Verify that the reference list complies with the guidelines for authors.
Authors’ answer: Thank you very much for your comment. We are prepared to make any
necessary adjustments based on further observations during the review process, should the
manuscript be accepted. We remain attentive to any additional requirements.

Round 2
Reviewer 1 Report
Comments and Suggestions for Authors
I am really grateful to re-review this manuscript. In my opinion, this manuscript can be published in current form.
Author Response
Authors’ answer: Thank you very much for your kind words and for taking the time to review our manuscript again. We deeply appreciate your valuable feedback and suggestions throughout the review process, which have significantly contributed to improving the quality of our work. We are truly grateful for your recommendation for publication.
Thank you once again for your support.

Reviewer 2 Report
Comments and Suggestions for Authors
I think that authors have addressed all my concerns, and I recomend it for publication.
Reviewer 3 Report
Comments and Suggestions for Authors
1. Please add clarification of the meaning of "lipids treatment".
2. It would be appropriate to mention that the validation of the method was supervised by one of the authors who focuses on diabetes and its complications.
3. Please define the abbreviation ATP from line 187.
4. Make sure figure captions are as descriptive as possible so that it is not necessary to read the main text to understand the intent and content.
5. It would be practical and systematic to name NO-DKD or YES-DKD for patients without DKD and yes-DKD on page 22. Throughout section and figures 12 and 13.
Author Response
1. Please add clarification of the meaning of "lipids treatment".
Authors’ answer: Thank you for your observation. We have addressed the concern by providing clear definitions for "lipids treatment" (LIPIDS.TX) and "hypertension treatment" (HA-TX) within the manuscript.
2. It would be appropriate to mention that the validation of the method was supervised by one of the authors who focuses on diabetes and its complications.
Authors’ answer: Thank you for your suggestion. A paragraph has been added to the discussion section highlighting that the study was supervised by experts. (lines: 561-564)
3. Please define the abbreviation ATP from line 187.
Authors’ answer: Thank you for your observation. We have defined the abbreviation ATP in line 187 as requested.
4. Make sure figure captions are as descriptive as possible so that it is not necessary to read the main text to understand the intent and content.
Authors’ answer: Thank you for your observation. We have revised the figure captions to make them more descriptive and self-explanatory.
5. It would be practical and systematic to name NO-DKD or YES-DKD for patients without DKD and yes-DKD on page 22. Throughout section and figures 12 and 13.
Authors’ answer: Thank you for your suggestion. We have implemented the changes as recommended. The terms "NO-DKD" and "YES-DKD" are now systematically used to refer to patients without DKD and with DKD, respectively, in the specified section and throughout Figures 12 and 13.
